# Finite element simulation study on vertical bearing characteristics of single pile with ram-compacted bearing sphere

**Lichao Bai[1], Shangwei Gong[1], Lina Xu[2], Xiaohong Bai[3], Zhanfang Huang[1] ***

**1** School of Civil Engineering and Geomatics, Shang Dong University of Technology, Zibo, Shandong Province, China, **2** School of Transportation Science and Engineering, Jilin Jianzhu University, Changchun, Jilin Province, China, **3** College of Civil Engineering, Taiyuan University of Technology, Taiyuan, Shanxi Province, China

* huangzhanfang@163.com

**Data Availability Statement:** All relevant data are within the paper and its Supporting Information files.

## Abstract

The pile with ram-compacted bearing sphere (*PRBS*)is a kind of special-shaped pile, the calculation formula of single pile bearing capacity stipulated in Chinese Standards JGJ/T 135–2018 is relatively simple, and the factors considered are not comprehensive enough. This article uses the finite element simulation software ABAQUS to simulate and calculate the compressive bearing characteristics of *PRBS*, and studies the changes in the vertical bearing characteristics of *PRBS* under different factors and working conditions (different pile lengths, pile diameters, and the diameters of ram-compacted bearing sphere (*RBS*)). The calculation results indicate that the *PRBS* still has a large axial force near the enlarged end of the pile bottom, and the *RBS* bears a large load. The vertical bearing capacity of the *PRBS* is mainly provided by the *RBS*, but the pile side friction still has a certain degree of influence on its bearing capacity. The maximum ratio of pile side frictional resistance to applied load can reach 18.41%. Compared with the ordinary pile, the bearing capacity of the *PRBS* is significantly improved. The ultimate bearing capacity of the *PRBS* with the *RBS* diameter of 1m is more than 5 times that of the ordinary pile under the same condition. Pile diameter has little influence on the bearing capacity of *PRBS*, while the change of *RBS* diameter has great influence on the bearing capacity of single pile. However, when the *RBS* diameter is too large, it is easy to cause the uplift of surrounding soil in the construction process and affect surrounding piles. Therefore, it is suggested that the optimal *RBS* diameter should be 800mm~1200mm. This study provides reference suggestions for the study of piles with ram-compacted bearing sphere.

## Introduction

With the rapid development of Chinese urban construction, the scale of engineering projects is expanding, and the requirements for foundation bearing capacity are becoming higher and higher. Therefore, it is necessary to develop different pile types to meet engineering needs. For

**Funding:** Lina Xu; Grant number:52008185; National Natural Science Foundation of China; https://www.nsfc.gov.cn/; The sponsors provided assistance in establishing numerical models.

**Competing interests:** The authors have declared that no competing interests exist.

this reason, researchers developed a new type of pile based on the expanded bottom pile and rammed expanded pile, namely pile with ram-compacted bearing sphere (*PRBS*) [1]. It combines the traditional pile foundation and foundation treatment technology, which can avoid soft soil layer, is a deep foundation treatment technology. It can provide high bearing capacity, fast construction speed, short construction period, good pile quality, avoid uneven settlement to a certain extent, and can deal with a large amount of construction waste during the construction process, significantly saving materials, low cost, and has good economic and environmental benefits, In line with the concept of green and sustainable development in China [2]. Since its invention, *PRBS* have been used as a typical expanded bottom pile in more than 200 cities and regions in China. They provide higher bearing capacity at lower costs and have been widely used in various foundation schemes for building facilities, is considered to be an efficient, reliable, and environmentally friendly pile foundation scheme [3]. And this method played an important role in the reconstruction work after the 2008 Sichuan earthquake, and was recognized by the government as one of the three main pile foundation technologies for Sichuan earthquake reconstruction [4,5].

At present, Chinese scholars have been conducting research on *PRBS* for more than 20 years, Qiu [6] found that the Q-s curve of the *PRBS* is a slowly changing type through finite element model analysis combined with on-site sampling and indoor model experiments. Wang [7] proved that the ram-compacted bearing sphere (*RBS*) can significantly improve the bearing capacity of pile foundation and reduce the foundation settlement through on-site static load test and finite Metacomputing. Zhang [8] studied the time effect and related factors on the bearing capacity of *PRBS* based on the three-phase theory of soil, and pointed out that the time effect has a significant impact on the bearing capacity of *PRBS*, and sufficient time should be left after the completion of pile formation to enhance the bearing capacity of *PRBS*. Yan [9] compared a variety of pile foundation treatment schemes for a project in Beijing and concluded that *PRBS* had great advantages in both economic and technical aspects. Zhou [10] conducted a theoretical analysis on the uplift bearing mechanism of *PRBS*, proposed a calculation formula suitable for the uplift bearing capacity of a single pile of *PRBS*, and verified its feasibility. Zhou [11] revealed the reinforcement mechanism during the formation process of *PRBS* from a theoretical perspective, providing a theoretical basis for establishing a calculation method for the vertical bearing capacity of *PRBS* considering the pile end compaction effect. Yang [12] determined through finite element analysis software that in the clay bearing layer, when the diameter of the *RBS* is 1–1.2 m, the optimal pile spacing of the *PRBS* is twice the diameter of the *RBS*; When the diameter of the *RBS* is greater than or equal to 0.8 m and less than 1 m, the optimal pile spacing of the *PRBS* is slightly greater than twice the diameter of the *RBS*.

Internationally, Du [13,14] used ABAQUS software to model a single pile with ram-compacted bearing sphere, taking into account the compaction effect during the construction process, and analyzed the bearing capacity of the pile and the characteristics of the soil, resulting in a more accurate and economical design solution. And a new finite element modeling cavity expansion method implemented in ABAQUS was proposed, which explored the driving mechanism of *PRBS* from the perspective of cavity expansion. The predicted ultimate bearing capacity results were verified through the static load test results of *PRBS* in historical cases, indicating that the finite element cavity expansion method is accurate and reliable. Kim [15,16] tested and analyzed the PHC pipe piles and EXT piles widely used in South Korea, and the results showed that compared with PHC piles, EXT piles with welded ring steel plates at the end increased the vertical bearing capacity by more than 20%, and could shorten the construction period and reduce the project cost. Gao [17] conducted a test study on 16 large-diameter hand-dug pile with enlarged bottom, and the results showed that increasing the

diameter of enlarged bottom had a more significant effect on improving the bearing capacity than increasing the pile diameter. Moayedi [18] provided an optimized artificial neural network prediction model for predicting the uplift capacity of bottom-expanded piles in dry sandy soil on the base of considering the main influencing factors, such as the diameter of bottom expanded tip, pile diameter, dip Angle of bottom expanded tip and burial depth rate. Kumar [19] proposed a prediction equation for the uplift bearing capacity of bottom-expanded piles based on the finite element simulation of the uplift bearing capacity of bottom-expanded piles in sandy soil, which has a good fit with the literature data.

To sum up, the *PRBS* technology can perfectly unify the quality, cost and environmental benefits, which will help realize the sustainable development of the construction industry. However, due to its short application time, the current Chinese Standards JGJ/T 135–2018 [20]still has some deficiencies in the calculation of the compression capacity of single pile. The standard calculation method is an empirical method, and the selection of two parameters in the calculation formula has great discreteness, so the design error is often large. Therefore, it is necessary to find a formula for calculating vertical ultimate bearing capacity of single *PRBS* with wide application and small error. Due to the limitations of high cost and time-consuming on-site experiments, this article uses finite element software ABAQUS to numerically simulate the bearing capacity of a single *PRBS* pile. It studies the changes in force and displacement of the pile under different pile lengths, pile diameters, and *RBS* diameters, analyzes and compares the effects of different conditions on the bearing capacity of a single *PRBS*.

## Construction process and bearing mechanism of *PRBS*

*PRBS* is a new type of countersunk tube ramming pile, composed of concrete pile body and composite ram-compacted bearing sphere (*RBS*). The *RBS* is a bearing body formed by heavy hammer compaction below the pile body. According to the soil properties of the strengthened soil, the *RBS* construction can be filled or not, so the *RBS* is divided into filling *RBS* and unfilled *RBS* [20]. The schematic diagram of the single pile structure of the *PRBS* is shown in Fig 1. The *RBS* consists of dry and hard concrete, compacted filling, compacted soil, and affected soil from inside to outside. In the construction process of *PRBS*, the column hammer is first used to tamping, followed by the back pressure of the casing to form a hole. When the design elevation is reached, batches of bricks, crushed stones, or cement sand mixture are filled into the hole and compacted. When the requirement for three blow penetration is reached, the dry hard concrete is filled and rammed again to form a composite *RBS* at the pile end. Then place the steel cage and pour the pile body [21]. The schematic diagram of the pile forming process of the *PRBS* is shown in Fig 2.

The bearing characteristics of *PRBS* are similar to those of expanded base piles and rammed expanded piles, which increase their bearing capacity by expanding the bearing area at the pile end, similar to multi-level unreinforced expansion foundations. The equivalent schematic diagram of the bearing capacity of the *PRBS* is shown in Fig 3. When the upper load of the structure acts on the top of the pile, the load is transmitted through the pile body to the reinforced *RBS*, which is composed of concrete, compacted filling material, and compacted soil, and the stress is gradually diffused and reduced. When transmitted to the bearing soil layer, the stress is less than the bearing capacity of the bearing layer foundation soil, thus meeting the requirements of the foundation bearing capacity. The compaction process of deep soil is the application innovation of *PRBS* technology and the stress diffusion principle is the stress principle of *PRBS* technology [22]. Comprehensive analysis shows that the bearing mechanism of *PRBS* has the following characteristics:

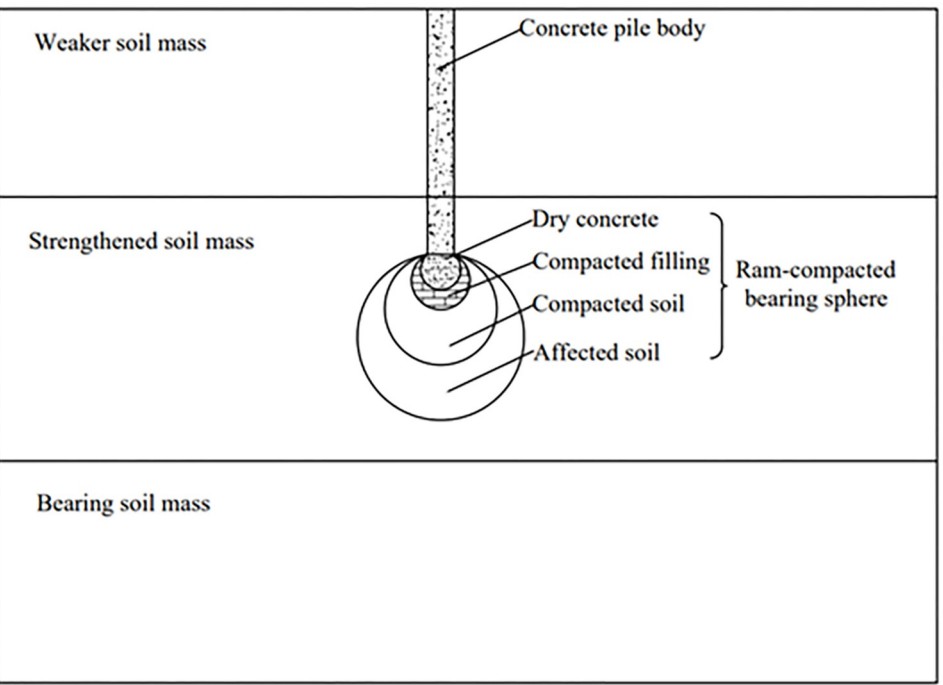

**Fig 1. The schematic diagram of the single pile structure of the *PRBS*.**

1. The load area of pile end gradually expands, and the stress diffused through the *RBS* reduces the stress concentration phenomenon of the soil at the pile end, and the pressure gradually decreases to the natural bearing layer.

2. The soil layer where the *RBS* is located is effectively compacted and strengthened, which greatly improves the bearing capacity, eliminates most of the compression deformation of the strengthened soil layer, and reduces the settlement of the building.

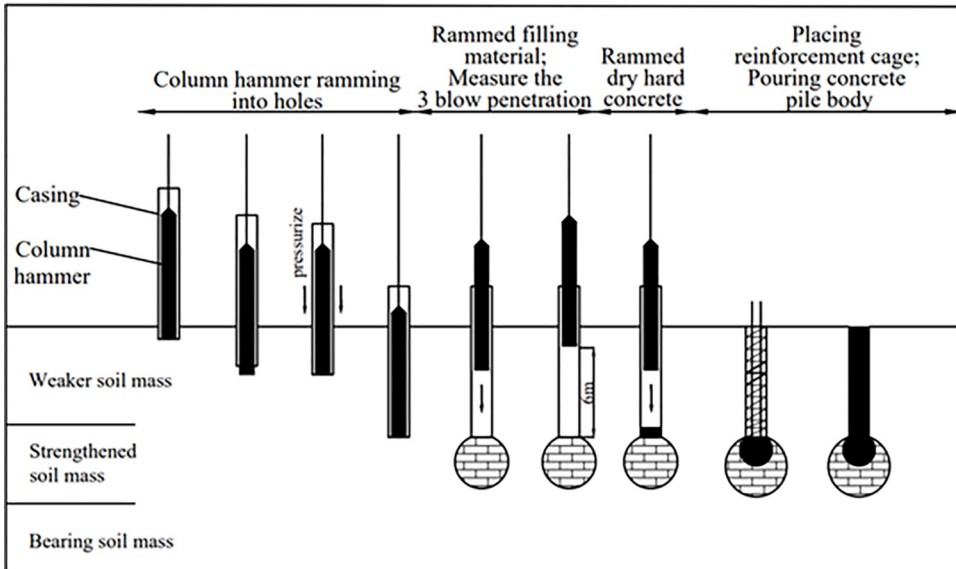

**Fig 2. The schematic diagram of the pile forming process of the *PRBS*.**

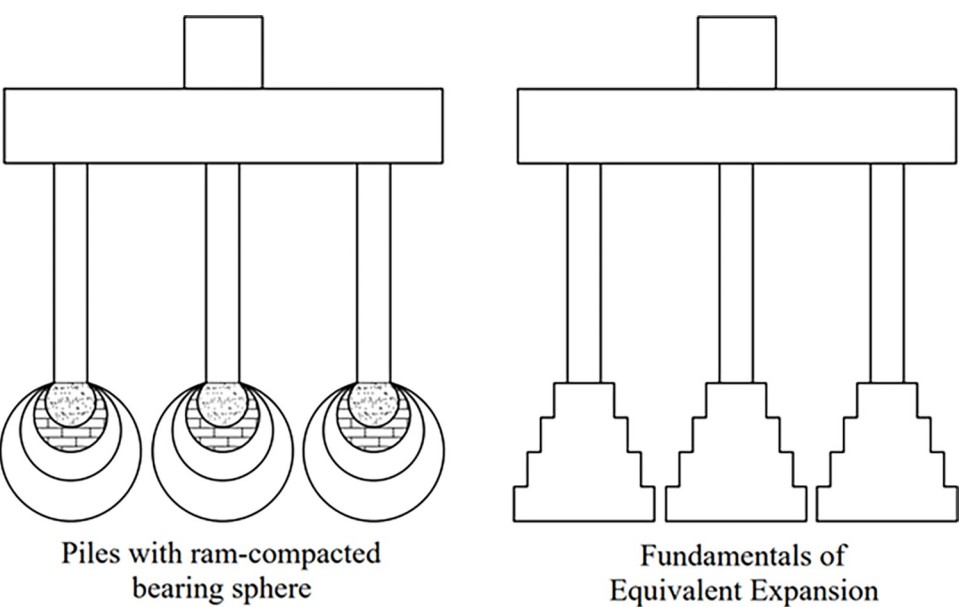

**Fig 3. The equivalent schematic diagram of the bearing capacity of the *PRBS*.**

According to the standard [20], in the preliminary design, for *PRBS* with pile length less than 30m, the characteristic value of vertical compression bearing capacity of single pile should be calculated according to Formula 1:

$$R_a = f_a \cdot A_e \tag{1}$$

In the equation: $f_a$ is the characteristic value of bearing capacity of *PRBS* bearing layer after modification (kPa), and the modification of bearing capacity shall be carried out according to the current code for Chinese Standards GB50007 [23]. $A_e$ is equivalent calculation area of *RBS* (m$^2$)

As the main bearing component of the *PRBS*, the bearing capacity of the *RBS* is related to the physical and mechanical properties of the soil at the pile end, and $A_e$ of the pile tip. The soil layer where the *RBS* is located is the reinforced soil layer. The initial *PRBS* is usually limited to viscous soil, silt, sandy soil, gravel soil, plain filled soil, and mixed filled soil as the reinforced soil layer. After continuous exploration of engineering practice in recent years, the strengthened soil layer has been extended to collapsible loess, residual soil, fully weathered rock and other soil or rock layers [20].

The value of the *RBS* equivalent calculated area is as follows: *RBS* construction is usually controlled by the three blows penetration degree and the amount of filling. At the initial stage of the application of *PRBS*, the value of $A_e$ only considers the two main influencing factors, namely the three blows penetration degree and the soil property of the reinforced soil layer. With the collection of engineering data and the improvement of the pore-forming ability of *PRBS*, the value of $A_e$ not only considers the above two factors, but also considers the influence of pile diameter in the Chinese Standards JGJ/T 135–2018.

## Introduction to finite element analysis

### Basic assumptions

Due to the complexity of numerical simulation, the following assumptions are made to simplify the calculation:

**Table 1. Main material parameters.**

| Model parameter | Elasticity parameter | | | Plastic parameters | |
|---|---|---|---|---|---|
| | | | | Mohr-Coulomb | |
| | Density (kg/m³) | E(MPa) | μ | φ(˚) | c(kPa) |
| Pile | 2600 | 30000 | 0.2 | - | - |
| Compacted soil | 2120 | 132 | 0.27 | 36 | 38 |
| Affected soil | 2030 | 63 | 0.3 | 27 | 27.8 |
| Weaker soil mass | 1800 | 12.8 | 0.43 | 16.7 | 14.7 |
| Strengthened soil mass | 1870 | 24 | 0.36 | 20 | 19.2 |
| Bearing soil mass | 1960 | 35.2 | 0.34 | 23.6 | 25.3 |

1. Assume that the soil is a continuous medium material, isotropic, and an ideal elastic-plastic material; The *PRBS* is an ideal continuous and homogeneous elastic body.

2. Considering the influence of initial crustal stress, the initial crustal stress balance was carried out on the whole model before loading.

3. The load is considered as static load.

4. Assume that the *RBS* is a regular sphere.

5. The influence of water change on soil in the model is not considered.

## Geometric model and related parameters

The size selection of pile is basically the same as the actual project. The model is mainly divided into two parts: the *PRBS* and the soil around the pile. When establishing the soil mass model, the horizontal width of the soil mass should be greater than or equal to 20 times the pile diameter (10m). The soil depth is about 2 times the pile length (20m). According to reference [6], under the condition of same pile diameter, for different diameters of the *RBS*, the range of the affected area of the *RBS* is close to one time the diameter of the compacted expansion body on the side and two times the diameter of the compacted expansion body at the bottom. At the same time, considering the simplification of finite element calculation, the soil mass is divided into three layers, which are 5m, 7m and 8m from top to bottom. The main material parameters are shown in Table 1, while other parameters are shown in Table 2, and the parameters of the pile are referenced from reference [24,25], and the parameters of the soil (such as internal friction angle) are referenced from reference [26]. Pile size and soil layer distribution are shown in Fig 4. According to the field sampling and laboratory model test mentioned in literature [27], we determined that the affected soil mass ranges from 2D under the ram-compacted bearing sphere (twice the diameter of the r ram-compacted bearing sphere) to 1D on the side of the ram-compacted bearing sphere (1 times the diameter of the ram-compacted bearing sphere), that is, its outer diameter is more than 3 times the diameter of the ram-compacted bearing sphere.

**Table 2. Other material parameters.**

| | Dilation Angle | Deviatoric eccentricity | Meridional eccentricity | Abs Plastic Strain | K0 |
|---|---|---|---|---|---|
| Soil1 | 0.1 | Calculated default | 0.1 | 0 | 0.71 |
| Soil2 | 0.1 | Calculated default | 0.1 | 0 | 0.67 |
| Soil3 | 0.1 | Calculated default | 0.1 | 0 | 0.57 |

## Constitutive model of pile and soil

In the study of pile-soil interaction, the accuracy and reliability of numerical simulation results are determined by selecting the appropriate constitutive model. The elastic modulus of concrete pile is much larger than that of soil, and failure rarely occurs when bearing vertical load. Moreover, the stress-strain relationship of pile body follows generalized Hooke's law. Therefore, the constitutive model of pile material in this paper adopts isotropic linear elastic model. Its expression is [25]:

$$
\begin{Bmatrix} \varepsilon_{11} \\ \varepsilon_{22} \\ \varepsilon_{33} \\ \gamma_{12} \\ \gamma_{13} \\ \gamma_{23} \end{Bmatrix} = \begin{bmatrix} 1/E & -v/E & -v/E & 0 & 0 & 0 \\ -v/E & 1/E & -v/E & 0 & 0 & 0 \\ -v/E & -v/E & 1/E & 0 & 0 & 0 \\ 0 & 0 & 0 & 1/G & 0 & 0 \\ 0 & 0 & 0 & 0 & 1/G & 0 \\ 0 & 0 & 0 & 0 & 0 & 1/G \end{bmatrix} \begin{Bmatrix} \sigma_{11} \\ \sigma_{22} \\ \sigma_{33} \\ \sigma_{12} \\ \sigma_{13} \\ \sigma_{23} \end{Bmatrix}
\tag{2}
$$

In the formula, $E$ is elastic modulus; $v$ is Poisson's ratio.

In this paper, the Mohr-Coulomb constitutive model is adopted for soil, and the Mohr-Coulomb plastic model is mainly applicable to granular materials under monotone loads, which is widely used in geotechnical engineering. The yield criterion of the Mohr-Coulomb model in ABAQUS is the shear failure criterion, which can also be set as the tensile failure criterion. The shear yield surface function in the Mohr-Coulomb model is:

$$
F = R_{mc}q - p \tan \varphi - c = 0
\tag{3}
$$

In the formula, $\varphi$ is the friction angle of the material, $0° \leq \varphi \leq 90°$; $c$ is the cohesion of the material; $\Theta$ is the extreme deflection Angle, defined as $\cos 3\Theta = \frac{r^3}{q^3}$; $r$ is the third deviator stress invariant $J_3$.

$R_{mc}(\Theta, \varphi)$ is calculated according to the following formula, which controls the shape of yield surface on the $\pi$ plane.

$$
R_{mc} = \frac{1}{\sqrt{3} \cos \varphi} \sin\left(\Theta + \frac{\pi}{3}\right) + \frac{1}{3} \cos\left(\Theta + \frac{\pi}{3}\right) \tan \varphi
\tag{4}
$$

The tensile failure criterion adopts the Rankine criterion:

$$
F_t = R_r(\Theta)q - p - \sigma_t = 0
\tag{5}
$$

Where, $R_r(\Theta) = (2/3) \cos 3\Theta$, $\sigma_t$ is the tensile strength, which varies with equivalent tensile plastic stress.

Fig 5 shows the shape of the Mohr-Coulomb yield surface on the meridian and $\pi$ planes.

## Contact settings

The pile-soil interaction belongs to the problem of contact between different media in solid mechanics. It is a typical nonlinear problem, which is manifested as the nonlinearity of materials and the nonlinearity of contact [28]. In this paper, the interaction mode of pile-soil is face to face. In the setting of contact pair, specify the contact between the pile and the soil and the pile with high stiffness is set as the master surface, and the soil is set as the slave surface.

The interaction of contact surface includes two parts: one is tangential action of contact surface, the other is normal action of contact surface. In this paper, the tangential effect of pile-soil contact is simulated by the method of Penalty function. The friction coefficient $\mu$ between

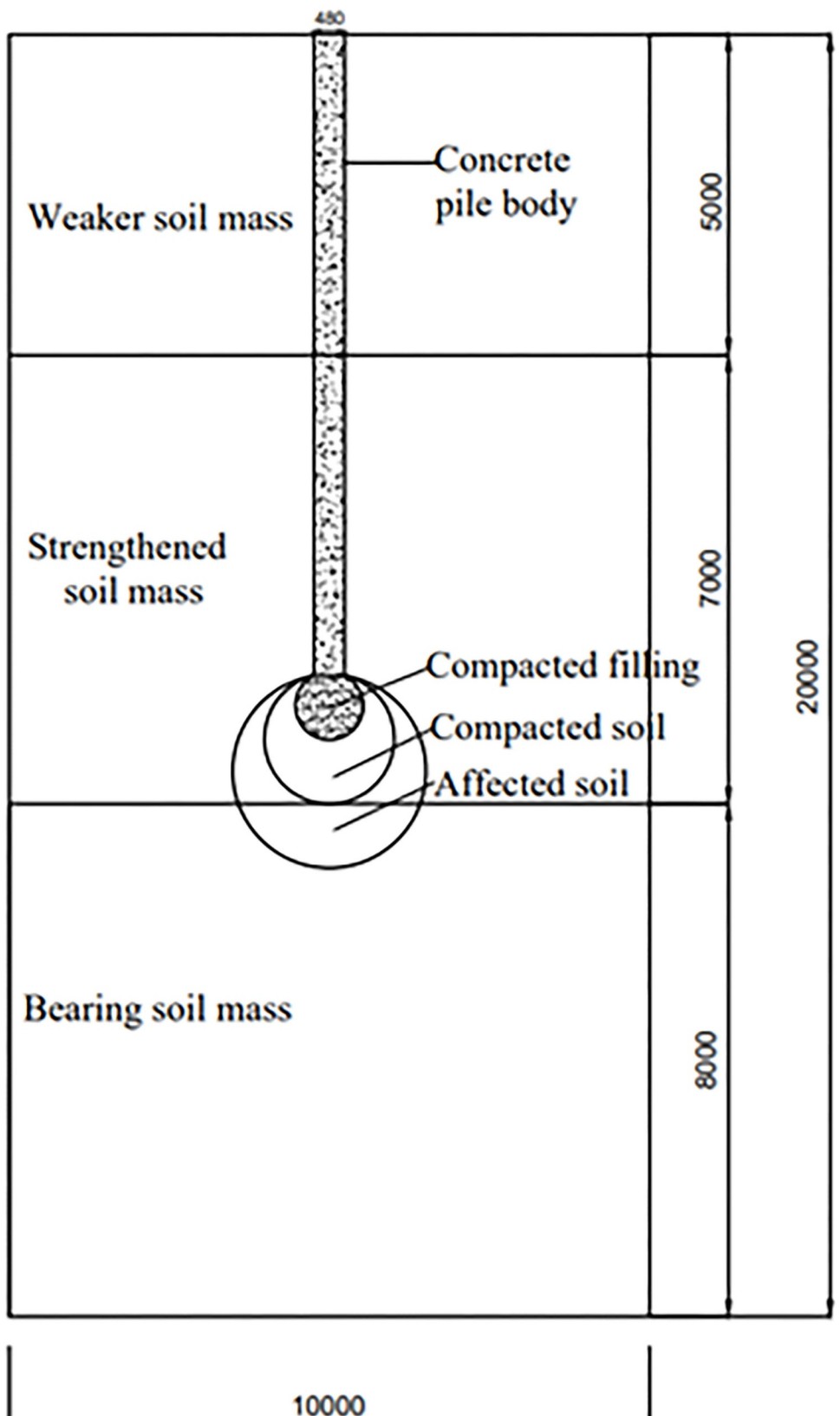

**Fig 4. Pile size and soil layer distribution (mm).**

pile-soil is calculated according to the following formula [29], this article takes 0.35:

$$\mu = \tan \delta \tag{6}$$

Where, $\delta$ is the friction angle of pile-soil contact surface, taking the value of $(0.75\sim1)\varphi$, $\varphi$ is the internal friction angle of soil mass.

The normal action of contact adopts the default "Hard" contact model of ABAQUS. The normal behavior of pile soil contact surface is very intuitive. If clearance exists between pile and soil, normal pressure cannot be transferred. It can only be transferred when pile and soil contact, and there is no limit on the pressure transferred during pile soil contact. That is, the normal compressive stress transmitted by the contact surface only exists in the contact compression of the object, if there is a clearance between the contact surface, there is no transfer force. This normal behaviour is defined in ABAQUS as "hard contact".

## Load and boundary conditions

We set a reference point at the pile top and couple it with the pile top plane. And then, we apply a point load to the reference point, which will evenly distribute the load on the pile top.

This paper mainly studies the bearing characteristics of *PRBS* under vertical loads. Therefore, the load transfer path is mainly that pile top bears vertical uniform pressure. Therefore, in the initial analysis step, the initial boundary conditions of all other surfaces in the overall model except pile top are set and activated, and displacement and corner constraints are applied, that is, horizontal and vertical fixed boundaries are adopted at the bottom of the model. The limited lateral displacement of the model is 0, and the top surface is a free boundary.

## Crustal stress balance

Crustal stress is a natural stress existing in the crust and not disturbed by engineering, also known as initial stress, absolute stress or original rock stress of rock mass, which is the fundamental force causing deformation and destruction of underground engineering. In the geotechnical engineering simulation involving the excavation of foundation pit, pile bearing capacity analysis and other problems, the initial crustal stress balance is a problem that must be considered, and the results of initial crustal stress balance will directly affect the accuracy of the calculation results [25]. The crustal stress balance is to make the model obtain a state with initial stress but no initial strain before simulation, and then apply the extracted internal force to the simulation, so that the simulation can achieve balance between the gravity and the applied internal force, so as to obtain accurate simulation results [25]. There are many methods for crustal stress balance, such as self-balance method, keyword definition method, ODB import method, etc. In the modeling process, we first used the Geostatic option provided by ABAQUS to do ground stress balance, but we found that the result of this method was not converging, so in this paper, multiple ODB import method is used to balance the crustal stress of the model.

The multiple ODB import method first establishes a "Static, General" analysis step, and then applies a force of -9.8N globally in the "Load" module for calculation. After the calculation is completed, return to the "Load" module and select the "Predefined field" command, the step selects "Initial", and then select the "Stress" in mechanical. The specification selects "From output database file" to select the calculation file, fill in the analysis step and increment step required for calculation completion, and then proceed with the calculation. After completing the calculation, change the "Static, General" analysis step mentioned earlier to the "Geostatic" analysis step, then return to the "Predefined field" command to change the file to a new file

and fill in the corresponding new analysis step and incremental step. At this point, the geostress balance is completed.

## Mesh partitioning

Mesh partitioning is an extremely important step in finite element analysis, and the number and quality of mesh partitioning directly affect the accuracy and scale of the calculation results. In order to appropriately reduce the calculation scale while ensuring the accuracy of the calculation, 0.11m should be applied to the pile body, compacted soil, and affected soil to make the mesh density relatively higher and the calculation results more accurate [30]. In order to appropriately reduce the scale of calculation and improve the efficiency of calculation, 0.8m is applied to soil mass. The use of appropriate element type in finite element analysis is helpful to better and more realistic simulation of the field situation. Since the *PRBS* is an irregular model, it is easy to have non-convergence when hexahedral modeling is adopted. Therefore, in this paper, a spatial four-node tetrahedron (C3D4) is used to model the single pile and soil mass. Although our research is related to axial symmetry, we did not choose to conduct axisymmetric simulation in this study. This decision is based on our detailed analysis and evaluation of the issue. We believe that under current models and conditions, the use of axisymmetric simulation is not necessary and may not have a significant impact on our main research objectives. In addition, 3D solid simulation are more intuitive, so this paper uses 3D solid simulation instead of axisymmetric simulation. The calculation result is convergent. The mesh division results are shown in Fig 6. The model has a total of 41635 nodes, with 235142 grids, which meet the requirements of engineering accuracy after calculation.

## Analysis of calculation results

### Stress nephogram and sedimentation nephogram

In order to study the mechanical and deformation characteristics of single pile with different types of *PRBS* under load, the influences of pile length, pile diameter and *RBS* diameter on the bearing capacity of single pile with *RBS* were analyzed. In this paper, a pile with a length of 10m, a diameter of 0.48m and a *RBS* diameter of 1m is taken as the reference pile. Load is applied to the top of the pile, and its stress nephogram and settlement nephogram are calculated, as shown in Fig 7. Parameters of different working conditions are shown in Table 3.

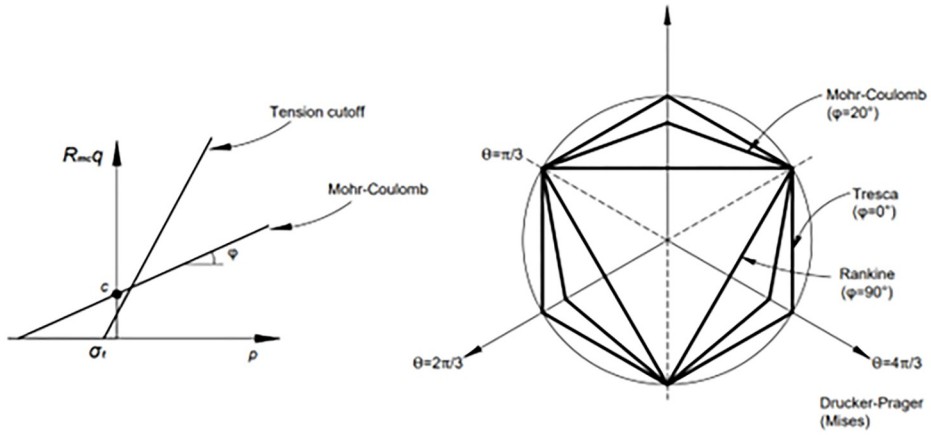

**Fig 5. Yield surface in the Mohr-Coulomb model [25].**

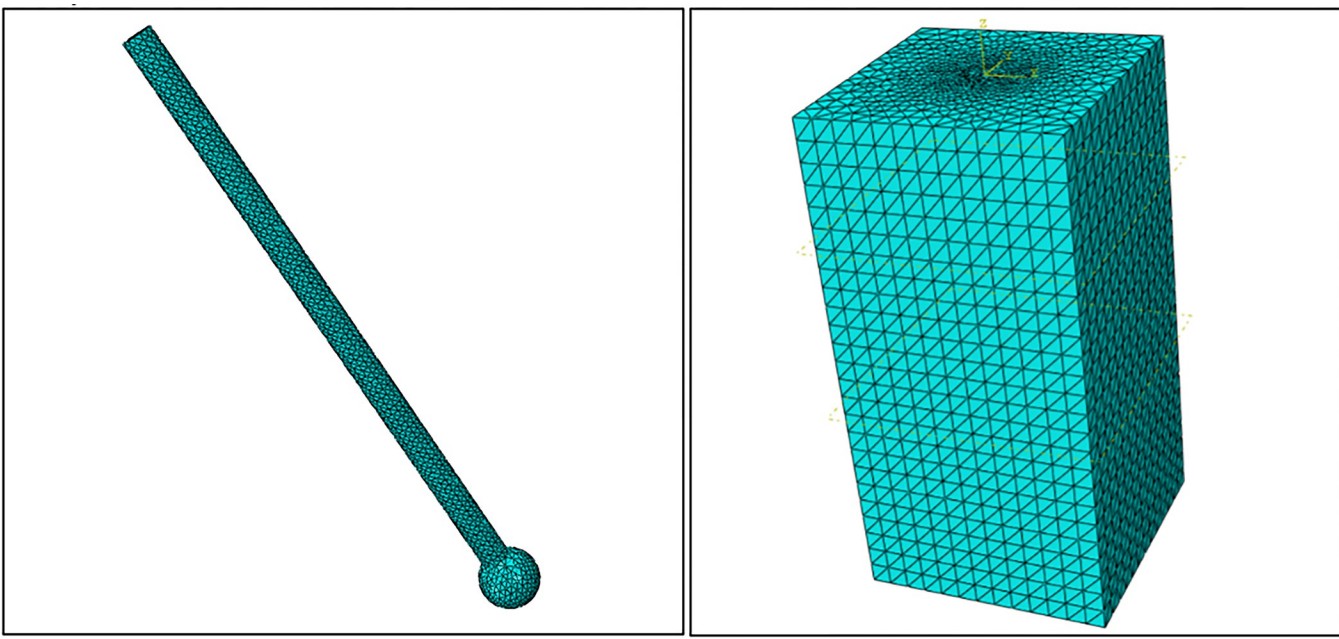

**Fig 6. Numerical model after mesh division.**

It can be seen from the settlement nephogram of the *PRBS* that the displacement at the bottom of the pile changes in a concave shape. The farther the distance from the bottom of the pile is, the less the soil around the pile is affected. It can be seen from the stress nephogram that the maximum stress occurs at the pile body and the stress concentration occurs. The stress

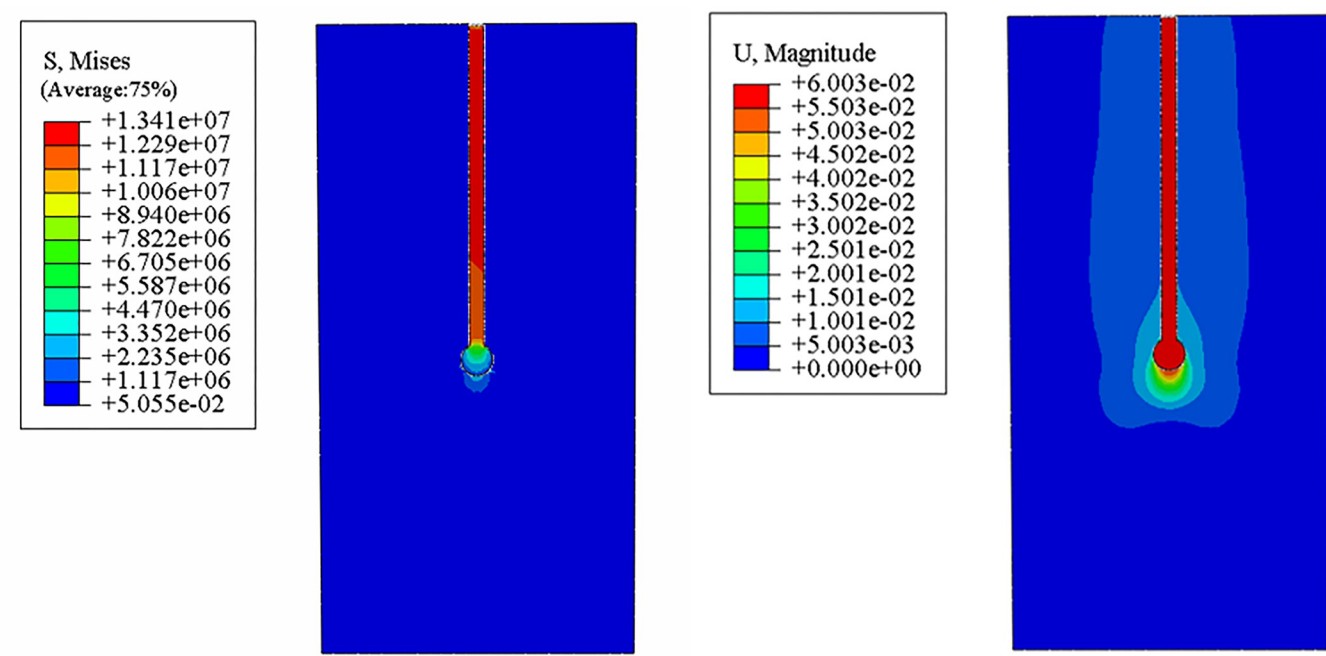

**Fig 7. Stress nephogram and settlement nephogram of reference pile.**

**Table 3. Model parameters of *PRBS* under different working conditions (m).**

| Number | ZC1 | ZC2 | ZC3 | ZC4 | ZC5 | ZC6 | ZJ1 | ZJ2 | ZJ3 | ZJ4 | ZJ5 | ZT1 | ZT2 | ZT3 | ZT4 | ZT5 | ZT6 |
|---|---|---|---|---|---|---|---|---|---|---|---|---|---|---|---|---|---|
| length | 4 | 6 | 8 | 10 | 12 | 14 | 10 | 10 | 10 | 10 | 10 | 10 | 10 | 10 | 10 | 10 | 10 |
| diameter | 0.48 | 0.48 | 0.48 | 0.48 | 0.48 | 0.48 | 0.44 | 0.48 | 0.52 | 0.56 | 0.6 | 0.48 | 0.48 | 0.48 | 0.48 | 0.48 | 0.48 |
| diameter of *RBS* | 1 | 1 | 1 | 1 | 1 | 1 | 1 | 1 | 1 | 1 | 1 | 0 | 0.6 | 0.8 | 1 | 1.2 | 1.4 |

at the joint of pile body and *RBS* decreases and the stress distribution is more uniform. The overall stress of *RBS* is less than that of pile body, indicating that the ramming body has the effect of diffusion stress. The vertical stress decreases with the pile body downward, indicating that part of the stress diffuses into the soil under the action of pile side friction, but the decreasing trend is not obvious, indicating that the *PRBS* is a typical end-bearing pile.

## Pile axial force and lateral friction resistance

Extract the axial force of different pile lengths from the ODB file, and obtain the distribution pattern of the axial force of the pile body extending the pile length under different load conditions, as shown in Fig 8.

It can be seen from Fig 8 that, under the same load, pile axial force gradually decreases with the increase of depth This is because when the vertical load appears on the pile top, the pile body will settle, resulting in the relative displacement between the pile and the soil body. In order to hinder the development of displacement, the soil on the side of the pile will produce upward friction resistance to the pile body. In the process of load transfer along the pile, the friction resistance needs to be overcome constantly, so the axial force decreases with the increase of depth. For general uniform cross-section piles, the axial force at the pile end is relatively small, but the *PRBS* still has a large axial force near the enlarged end of the pile bottom, and the *RBS* bears a large load, exhibiting the bearing characteristics of end-bearing piles.

The calculation of the bearing capacity of the *PRBS* in the standard only considers the pile end resistance and does not consider the pile side friction resistance. However, as shown in Fig 8, there is a certain slope in the axial force distribution curve of the pile body, indicating that the side friction resistance has a certain impact on the bearing capacity of the *PRBS*. The ratio of pile side friction resistance to ultimate bearing capacity under various working conditions is shown in Table 4. It can be seen from the table that as the pile length increases, the pile side friction resistance at the ultimate load also increases, and the proportion of pile side friction resistance also increases accordingly, with the maximum proportion reaching 18.41%.

Generally speaking, the bearing capacity of pile is provided by pile side friction resistance and pile end resistance, even if the pile end bearing capacity is the main bearing mode, there will be pile side friction. And even where the contact quality is poor, side friction can provide additional support by increasing the friction between the pile and the soil, thereby increasing the total load carrying capacity of the pile. So we believe that considering pile side friction is important.

Pile side friction under different loads under different working conditions is extracted, as shown in Table 5. By comparing the lateral friction resistance obtained with the applied load, the percentage of pile lateral friction resistance in the applied load under each working condition is obtained, as shown in Table 6.

It can be seen from Table 5 that under different working conditions, the required load for the complete development of pile side friction is different. The percentage of pile side friction in the applied load shows a trend of first increasing and then decreasing with the increase of load. This is because when the load is relatively low, the side friction is fully developed and the

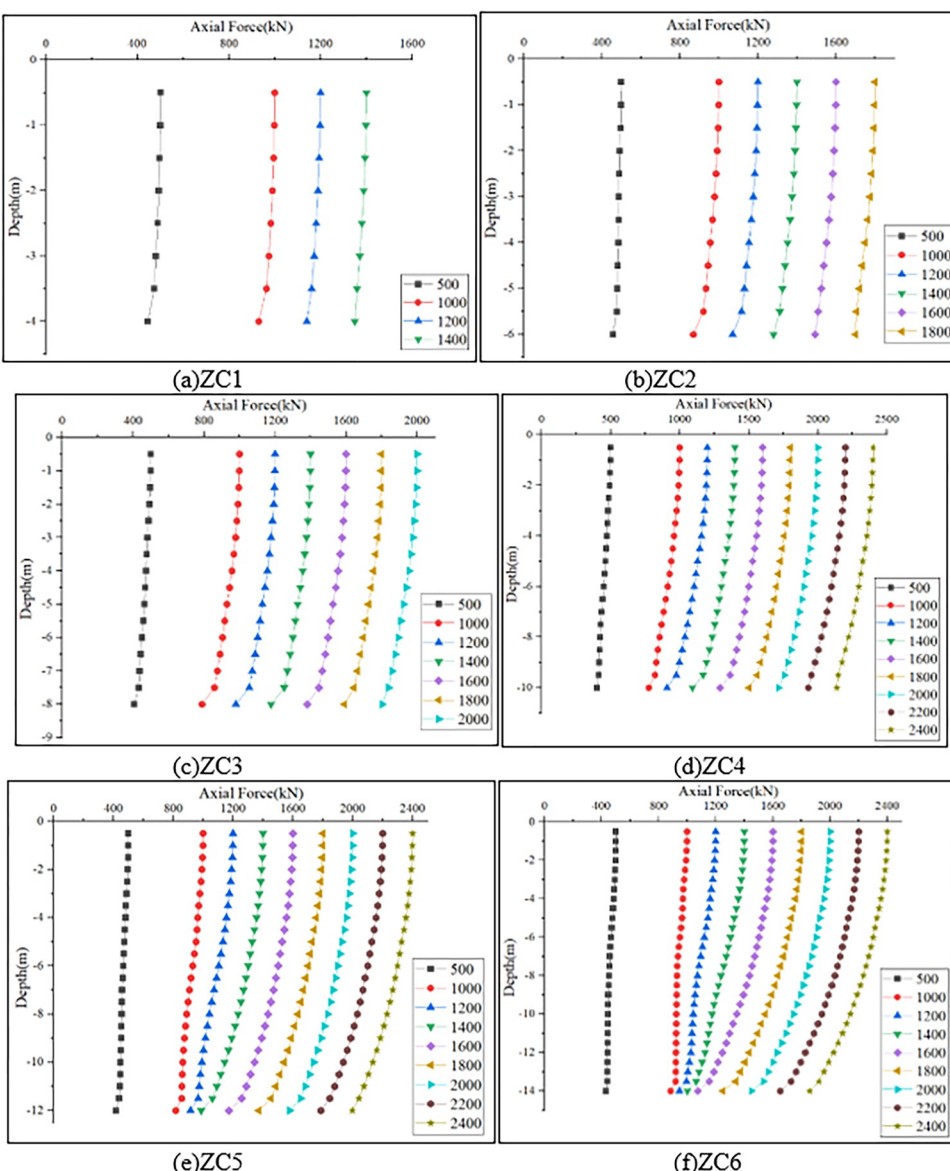

**Fig 8. Pile axial force distribution curves under different load conditions.**

proportion is relatively high. As the load increases, the side friction is fully developed and basically does not change, so the proportion decreases accordingly. Note that when the load is low, the pile side friction first increases and then decreases with the increase of pile length. However, when the load exceeds 1400kN, the pile side friction gradually increases with the increase of pile length. Therefore, for practical engineering, when the upper load is small, it is recommended that the pile length be between 6m and 10m, and it is not recommended to be too

**Table 4. Pile side friction and its percentage when the ultimate bearing capacity is reached in each working condition.**

|  | ZC1 | ZC2 | ZC3 | ZC4 | ZC5 | ZC6 |
|---|---|---|---|---|---|---|
| Pile side friction (kN) | 53 | 109.23 | 205.15 | 296.41 | 450.44 | 615.92 |
| percentage (%) | 3.60% | 6.04% | 9.51% | 11.35% | 15.19% | 18.41% |

**Table 5. Pile side friction under different load conditions (kN).**

| Number | 500kn | 1000kn | 1200kn | 1400kn | 1600kn | 1800kn | 2000kn | 2200kn | 2400kn |
|--------|-------|--------|--------|--------|--------|--------|--------|--------|--------|
| ZC1 | 55.62 | 69.90 | 60.00 | 51.00 | | | | | |
| ZC2 | 43.52 | 130.80 | 130.00 | 120.00 | 107.00 | 99.00 | | | |
| ZC3 | 94.30 | 211.80 | 220.70 | 223.00 | 219.00 | 209.00 | 197.00 | | |
| ZC4 | 98.60 | 223.20 | 290.90 | 308.00 | 307.00 | 299.00 | 285.00 | 269.00 | 262.00 |
| ZC5 | 83.00 | 183.19 | 284.45 | 413.14 | 427.87 | 428.19 | 422.00 | 403.00 | 386.00 |
| ZC6 | 68.40 | 149.58 | 254.50 | 399.00 | 524.97 | 551.00 | 552.00 | 549.00 | 542.00 |

long to avoid stress concentration caused by the *RBS* bearing large loads; When the foundation needs to withstand large loads, it is recommended to increase the pile length to fully exert the effect of pile side friction resistance.

From Table 6, it can be seen that the maximum values of the percentage of lateral frictional resistance of ZC1 to ZC6 piles to the applied load are 11.12%, 13.08%, 21.18%, 24.24%, 29.51%, and 32.81%, respectively, With the increase of pile length, the ratio of friction resistance also increases, and when the pile length is 14m, the maximum value of pile lateral friction resistance has reached 32.81%, indicating that pile lateral friction has a great degree of influence on the bearing capacity of *PRBS*, which cannot be ignored. When the pile length reaches a certain depth, The calculation of bearing capacity of *PRBS* should consider not only pile end resistance but also pile side friction resistance.

## The effect of pile length on the compressive bearing performance of a single *PRBS*

The length of the pile body is an important factor affecting the bearing performance of the *PRBS*. Under the condition of ensuring the same diameter of the pile body and *RBS*, the influence of gradually changing the pile length on the vertical bearing performance of the *PRBS* is studied. Establish models with pile lengths of 4m, 6m, 8m, 10m, 12m, and 14m for analysis and calculation, and obtain the ultimate bearing capacity of the pile under different pile length conditions as shown in Table 7.

Fig 9 shows the Q-s curve of the *PRBS* under different pile length conditions. It can be seen from the figure that the six groups of single *PRBS* models all show a slow variation characteristic, with no obvious steep drop point. The longer the pile length, the higher the bearing capacity of the *PRBS*, mainly because the increase of the pile length changes the pile soil contact area, so that the pile side friction is exerted more and more, and the bearing capacity increases accordingly. With the increase of pile length, the downward trend of Q-s curve gradually slows down, and the settlement decreases under the same load. Taking the pile top load of 1000kN as an example, the pile top settlement from ZC1 to ZC6 is 20.88mm, 11.86mm, 7.77mm,

**Table 6. Percentage of pile side friction in applied load under different working conditions (%).**

| Number | 500kn | 1000kn | 1200kn | 1400kn | 1600kn | 1800kn | 2000kn | 2200kn | 2400kn |
|--------|-------|--------|--------|--------|--------|--------|--------|--------|--------|
| ZC1 | 11.12% | 6.99% | 5.00% | 3.64% | | | | | |
| ZC2 | 8.70% | 13.08% | 10.83% | 8.57% | 6.69% | 5.5% | | | |
| ZC3 | 18.86% | 21.18% | 18.39% | 15.93% | 13.69% | 11.61% | 9.85% | | |
| ZC4 | 19.72% | 22.32% | 24.24% | 22.00% | 19.19% | 16.61% | 14.25% | 12.23% | 10.92% |
| ZC5 | 16.60% | 18.32% | 23.70% | 29.51% | 26.74% | 23.79% | 21.10% | 16.79% | 12.87% |
| ZC6 | 13.68% | 14.96% | 21.21% | 28.50% | 32.81% | 30.61% | 27.60% | 24.95% | 22.58% |

**Table 7. Ultimate bearing capacity of *PRBS* under different pile length conditions (kN).**

| Number | ZC1 | ZC2 | ZC3 | ZC4 | ZC5 | ZC6 |
|---|---|---|---|---|---|---|
| Ultimate bearing capacity | 1472.05 | 1808.78 | 2156.28 | 2610.51 | 2965.22 | 3346.2 |

5.27mm, 5.16mm and 5.09mm, respectively. According to the standard [20], when Q-s is a slowly changing curve, the vertical ultimate bearing capacity of single pile should be determined by taking the load value corresponding to the total settlement of pile top of 60mm. Therefore, the ultimate bearing capacity of single pile from model ZC1 to ZC6 is 1472.05kN, 1808.78kN, 2156.28kN, 2610.51kN, 2965.22kN and 3346.2kN, respectively. Compared with ZC1, the bearing capacity of ZC2 to ZC6 increased by 22.87%, 46.48%, 77.34%, 101.43% and 127.32%, respectively. When the pile length increases from 8m to 10m, the increase in bearing capacity is relatively large. This is because the bearing soil layer has changed. When the pile length is 10m, the bearing soil layer becomes more solid. At this moment, not only does the pile side friction resistance increase, but also the pile end resistance increases to a certain extent, so the increase in bearing capacity is relatively large. Therefore, in the design and construction process of *PRBS*, it is necessary to fully consider the impact of the bearing soil layer on the bearing performance of the *PRBS*, and choose a soil layer with better soil quality as the bearing layer to fully utilize the improvement of the bearing capacity brought by the *RBS*.

## Influence of *RBS* diameter on compressive bearing capacity of single *PRBS*

*PRBS* is a typical end-bearing pile, which is similar to extending the force of foundation, so the diameter of *RBS* has an important effect on its bearing performance. With other conditions

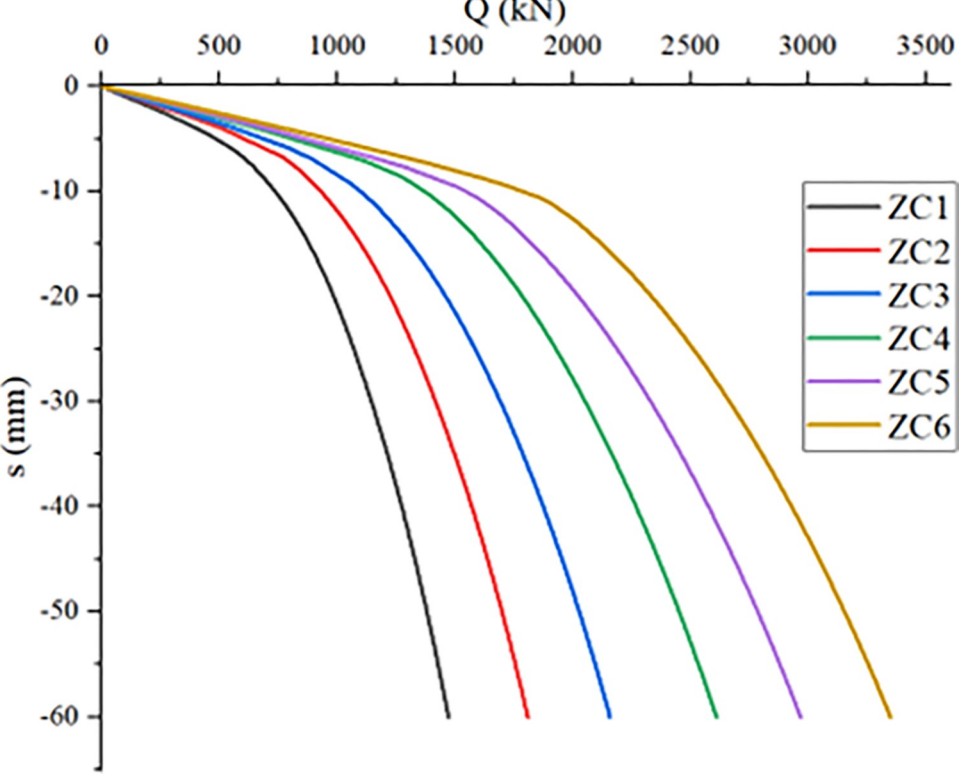

**Fig 9. Q-s curve of piles under different pile lengths.**

**Table 8. Ultimate bearing capacity of *PRBS* under different *RBS* diameters (kN).**

| Number | ZT1 | ZT2 | ZT3 | ZT4 | ZT5 | ZT6 |
|---|---|---|---|---|---|---|
| Ultimate bearing capacity | 498.53 | 724.88 | 1848.52 | 2610.51 | 3419.10 | 4303.15 |

unchanged, simulation was carried out on single piles with *RBS* diameters of 0.6m, 0.8m, 1.0m, 1.2m, 1.4m and without *RBS* respectively, and the ultimate bearing capacity of piles under different *RBS* diameters was obtained, as shown in Table 8. Q-s curves under different working conditions were shown in Fig 10.

It can be seen from Table 8 that with the increase of the diameter of the *RBS*, the bearing capacity increases significantly. This is because with the increase of the diameter of the *RBS*, the range and degree of the soil reinforcement at the pile end also increases gradually. The *PRBS* is a pile type dominated by the pile end resistance and supplemented by the pile side resistance. As the diameter of the *RBS* increases, the effective area of pile end resistance also increases, and the bearing capacity also increases. Observing the Q-s curve, the slope of the curve is small in the initial stage of loading, which is approximately linearly increasing. At this time, the load is mainly borne by the pile side resistance. As the load increases, the pile side resistance reaches the limit state, and the pile end resistance begins to play a dominant role. There are differences in the Q-s curve of piles with different *RBS* diameters. The larger the diameter of *RBS*, the smoother the Q-s curve, the greater the bearing capacity of the pile, and the smaller the settlement.

Due to the small diameter of the ordinary non-*RBS* pile and the unstable bearing soil layer, the pile tip penetrates into the bearing layer during failure, exhibiting a steep descent curve.

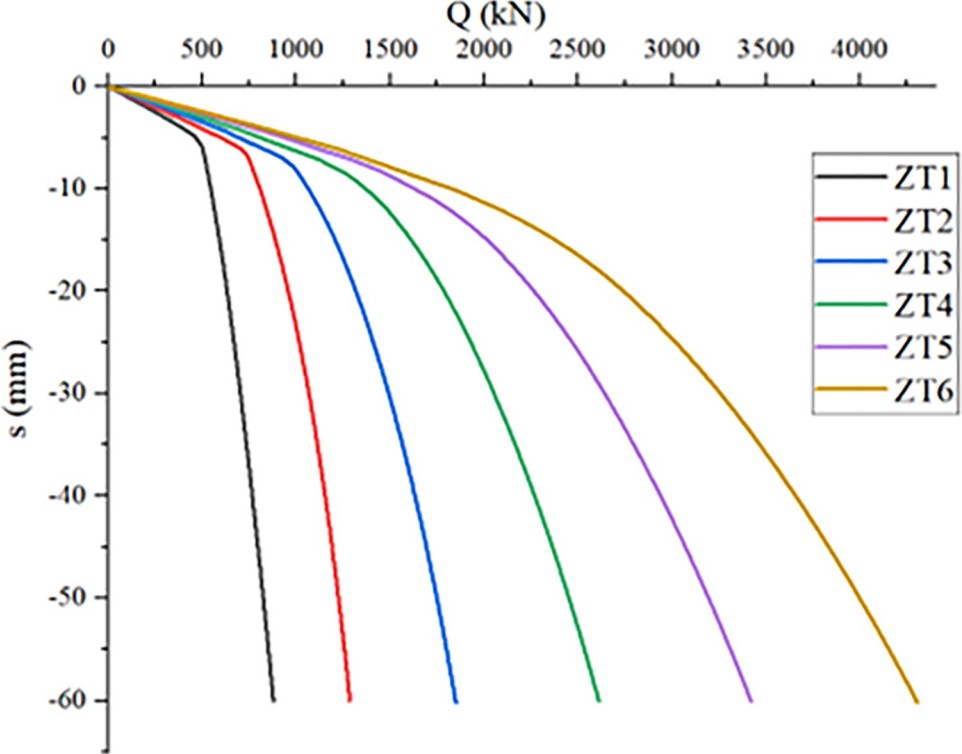

**Fig 10. Q-s curves of single pile with different *RBS* diameters.**

**Table 9. Ultimate bearing capacity of _PRBS_ under different pile diameters (kN).**

| Number | ZJ1 | ZJ2 | ZJ3 | ZJ4 | ZJ5 |
|---|---|---|---|---|---|
| Ultimate bearing capacity | 2559.0 | 2610.51 | 2655.70 | 2683.99 | 2719.22 |

For steep drop curves, the load corresponding to the point where the Q-s curve undergoes a significant drop is taken as its ultimate load, so the bearing capacity of ZT1 is taken as 498.53kN. Due to the small difference between the _RBS_ diameter and pile diameter of ZT2, as well as the unstable bearing soil layer, penetration failure occurs, and its Q-s curve also shows a steep drop, with a bearing capacity of 724.88kN. Compared with ZT1, the bearing capacity of ZT3 to ZT6 increases by 270.79%, 423.64%, 585.84% and 763.17%, respectively. It can be seen that the bearing capacity of pile is greatly influenced by the diameter of _RBS_. However, considering that an increase in the diameter of the _RBS_ requires a greater amount of hammering energy from the column hammer, which will increase the difficulty and cost of construction. When the diameter of the rammed expansion body increases to 1.4m, it is easy to cause uplift of the surrounding soil during the construction process, causing adverse effects on adjacent piles that have already been completed, leading to significant construction uncertainty. This pile type is rarely used in engineering, so it is recommended that the optimal _RBS_ diameter be 800mm~1200mm.

### Influence of pile diameter on compressive bearing performance of single _PRBS_

Under the same conditions of controlling the length of the pile body, _RBS_ diameter, and soil conditions, the influence of pile diameter on the vertical bearing capacity of the _PRBS_ is studied by gradually changing the pile diameter. The ultimate bearing capacity of the _PRBS_ under different pile diameter conditions is shown in Table 9, and its Q-s curve is shown in Fig 11.

As can be seen from the figure, when other conditions remain unchanged, the larger the pile diameter, the smaller the pile tip settlement and the larger the bearing capacity, but the pile tip settlement and bearing capacity change is not significant, the ultimate bearing capacity of a single pile increased from 2559kN to 2719.22kN, an increase of only 6.26%. This is because the pile end resistance is the main bearing capacity of the _PRBS_, and increasing the pile diameter cannot significantly improve the pile end resistance. Therefore, for general engineering, it is difficult to significantly improve the vertical bearing capacity by increasing the pile diameter, so it is not recommended to change the vertical bearing capacity of _PRBS_ only by increasing the pile diameter.

### Comparison and verification of numerical simulation results and standard formula bearing capacity

Substitute the data of the simulated working conditions into Formula (1) to obtain the characteristic value of the compressive bearing capacity of a single pile. Among them:

$$f_a = f_{ak} + \eta_b \gamma (b - 3) + \eta_d \gamma_m (d - 0.5) \tag{7}$$

Where, $f_{ak}$ is the characteristic value of foundation bearing capacity (kPa). In this paper, the characteristic value of foundation bearing capacity of silty clay is 150; $\eta_b$ is the width correction coefficient of foundation, and the width correction coefficient of _PRBS_ is 0; $\eta_d$ is the correction coefficient of foundation depth. In this paper, for the viscous soil with both e and IL less than 0.85, 1.6 is taken according to the specification [23]; $\gamma_m$ is the weighted average weight of soil

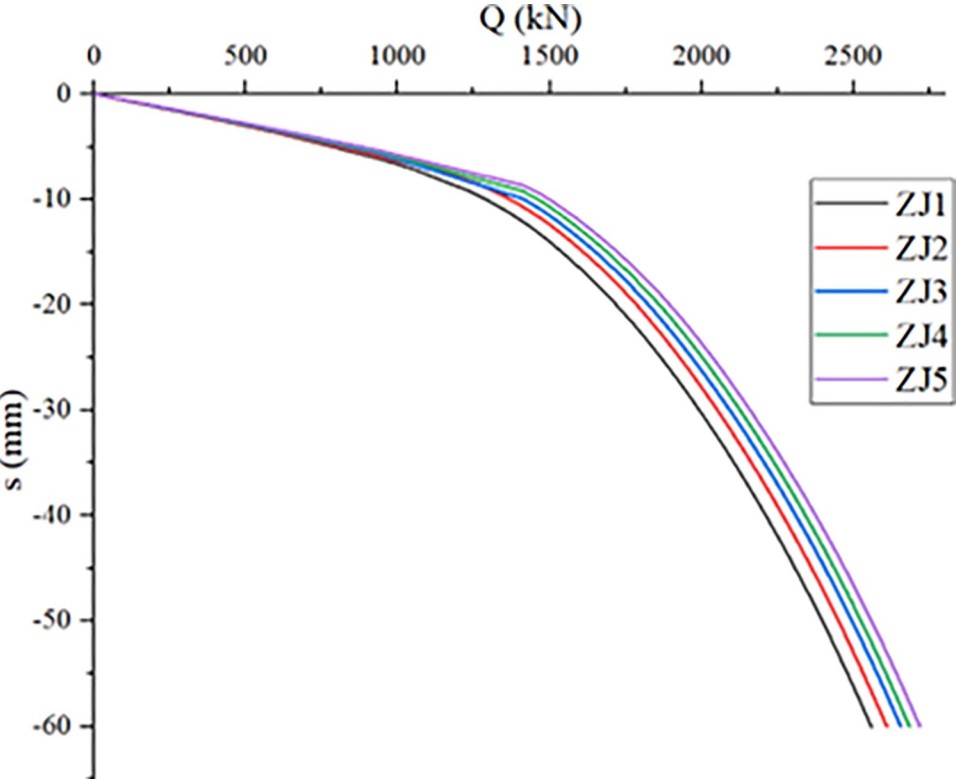

**Fig 11. Influence of pile diameter on bearing capacity of *PRBS*.**

above the foundation bottom (kN/m3), this paper takes 18 kN/m3; $d$ is the depth of foundation burial (m).

According to the standard [20], for the *PRBS* in the cohesive soil $0.25 \leq I_L \leq 0.75$ and with a pile diameter of 450mm~500mm, the equivalent calculated area $A_e$ of the *RBS* is 2.2~2.8. When the pile diameter is 350mm~450mm, $A_e$ should be multiplied by 0.85~0.95, and when the pile diameter is 500mm~800mm, $A_e$ should be multiplied by 1.1~1.3.

So for ZC1, its $R_a = f_a \cdot A_e = [150 + 1.6 \times 18 \times (5 - 0.5)] \times 2.5 = 699\text{kN}$

Similarly, the vertical bearing capacity characteristic values of single pile under other working conditions are obtained.

According to the ultimate bearing capacity of single pile calculated by finite element method, the characteristic value of vertical compression bearing capacity of single *PRBS* specified in the standsrd[20] is calculated as follows:

$$R_a = Q_{uk}/K \tag{8}$$

In the formula, $Q_{uk}$ is the standard value of ultimate bearing capacity of single *PRBS* under vertical compression (kN); $K$ is the safety factor, taking $K = 2$.

The ultimate bearing capacity of each working condition obtained by finite element simulation was substituted into Formula (8) to calculate the characteristic value of the compressive bearing capacity of single pile in each working condition, and compare them with the results calculated by Formula (1), as shown in Table 10. The comparison between the bearing capacity of the RBS bearing and the standard is shown in Table 11.

It can be seen from the Table 10 that the characteristic values of vertical compression bearing capacity of single *PRBS* under different working conditions obtained by numerical

**Table 10. Comparison between numerical simulation results and standard calculation results (kN).**

| Number | ZC1 | ZC2 | ZC3 | ZC4 | ZC5 | ZC6 | ZJ1 | ZJ2 | ZJ3 | ZJ4 | ZJ5 | ZT1 | ZT2 | ZT3 | ZT4 | ZT5 | ZT6 |
|---|---|---|---|---|---|---|---|---|---|---|---|---|---|---|---|---|---|
| standard calculation | 699 | 843 | 987 | 1131 | 1275 | 1419 | 1087 | 1156 | 1287 | 1317 | 1348 | - | - | - | - | - | - |
| numerical simulation | 736 | 904 | 1078 | 1305 | 1482 | 1673 | 1279 | 1305 | 1328 | 1342 | 1359 | 249 | 362 | 924 | 1305 | 1710 | 2152 |

**Table 11. Comparison between the bearing capacity of the RBS bearing and the standard (kN).**

| Number | ZC1 | ZC2 | ZC3 | ZC4 | ZC5 | ZC6 |
|---|---|---|---|---|---|---|
| Ultimate bearing capacity | 1472.05 | 1808.78 | 2156.28 | 2610.51 | 2965.22 | 3346.2 |
| Side friction resistance | 53 | 109.23 | 205.15 | 296.41 | 450.44 | 615.92 |
| Pure RBS bearing capacity without friction | 1419.05 | 1699.55 | 1951.13 | 2314.1 | 2514.78 | 2730.28 |
| Characteristic value of pure RBS bearing contribution bearing capacity | 709.525 | 849.775 | 975.565 | 1157.05 | 1257.39 | 1365.14 |
| Standardized calculation of bearing capacity characteristic values | 699 | 843 | 987 | 1131 | 1275 | 1419 |

simulation are all greater than those obtained by the standard, indicating that the standard define the bearing capacity of *PRBS* relatively conservatively. For piles less than 15m in length, it is suggested to multiply the characteristic values of its compression bearing capacity by a correction factor of 1.05~1.2. Since the diameter of the *RBS* cannot be directly determined in on-site construction and can only be estimated by the amount of filler, the relationship between the diameter of the *RBS* and the equivalent calculated area $A_e$ of the *RBS* is not recorded in the standard. However, the finite element calculation shows that the bearing capacity of the *PRBS* has a great relationship with the diameter of the *RBS*, so the relationship between the diameter of the *RBS* and the equivalent calculated area of the *RBS* needs to be further studied.

From Table 11, we note that the characteristic values of the RBS bearing capacity of ZC3 to ZC6 are slightly lower than those calculated by the standard. This is because the standard calculation is an empirical calculation that takes into account the uncertainty brought by various soil parameters. The characteristic values of the RBS bearing capacity are directly obtained through numerical simulation and theoretical analysis, and their calculation methods are more direct and accurate. Therefore, in general, it is normal for the characteristic value of the bearing capacity of the RBS bearing to be slightly lower than the calculated characteristic value of the bearing capacity according to regulations.

## Conclusions and suggestions

Finite element software ABAQUS was used for numerical simulation of *PRBS*, and the bearing capacity under different conditions was compared, and the analysis was concluded as follows:

1. *PRBS* shows the bearing characteristics of end-bearing pile, but pile side friction has an undeniable effect on the bearing capacity of *PRBS*. The maximum ratio of pile side frictional resistance to applied load can reach 18.41%. The calculation of bearing capacity of *PRBS* should not only consider the pile end resistance, but also consider the play of pile side friction resistance.

2. In the case of the same pile length and *RBS* diameter, as the pile diameter increases, the bearing capacity of the *PRBS* increases, but it is not obvious. The influence of the pile diameter on the bearing capacity of the *PRBS* is relatively small. Therefore, it is not recommended for general engineering to only change the vertical bearing capacity of the *PRBS* by increasing the pile diameter.

3. Under the same conditions of pile length and pile diameter, an increase in the diameter of the *RBS* will have a significant impact on the bearing capacity of the *PRBS*. Compared to the ordinary pile type without a *RBS*, the bearing capacity of the *PRBS* with a diameter of 1m has increased by 423.64%, and the bearing capacity has been significantly improved. However, when the diameter of the *RBS* is too large, it can easily cause uplift of the surrounding soil during the construction process, causing adverse effects on the adjacent piles that have already been completed, and bringing significant construction uncertainty, Therefore, it is recommended that the optimal *RBS* diameter be 800mm~1200mm.

4. However, this paper only analyzes and studies the bearing characteristics of *PRBS* through numerical simulation method, and lacks comparison with certain engineering tests. And this paper only the compression at the top of the pile has been analyzed, and in future research, the uplift resistance of the *PRBS* and its bearing characteristics under earthquake action can be considered

## Supporting information

**S1 Data. The values used to build graphs.**
(ZIP)

## Author Contributions

**Conceptualization:** Shangwei Gong.

**Formal analysis:** Shangwei Gong.

**Funding acquisition:** Lina Xu.

**Methodology:** Lichao Bai.

**Resources:** Lina Xu.

**Supervision:** Xiaohong Bai, Zhanfang Huang.

**Writing – original draft:** Lichao Bai.

**Writing – review & editing:** Xiaohong Bai, Zhanfang Huang.

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
