## [Decision Letter · Decision Letter 0]

10 Jul 2023

PONE-D-23-17951Finite Element Simulation Study on Vertical Bearing Characteristics of Single Pile with Ram-Compacted Bearing SpherePLOS ONE

Dear Dr. Huang,

Thank you for submitting your manuscript to PLOS ONE. After careful consideration, we feel that it has merit but does not fully meet PLOS ONE’s publication criteria as it currently stands. Therefore, we invite you to submit a revised version of the manuscript that addresses the points raised during the review process.

Overall, the authors must bring discussions and insights which are different from the ones already discussed in the works [3] and [13]. Without that, it is impossible to check the full originality of the present paper. Besides, since considering the friction contribution is deeply related to the quality of the soil-pile contact, it is important to show how different values of the penalty friction impact the final bearing capacity. This would make the readers more aware of how the Chinese Standard performs in practice.

We look forward to receiving your revised manuscript.

Kind regards,

Luan Carlos de Sena Monteiro Ozelim, D.Sc.

Academic Editor

PLOS ONE

“Lina Xu；Grant number:52008185； National Natural Science Foundation of China; https://www.nsfc.gov.cn/ ; The sponsors provided assistance in establishing numerical models.”

Additional Editor Comments:

Please address both reviewer's comments. Besides, I would add certain issues which also need clarification:

a) About the ABAQUS model:

- How was the load applied? Were increments of load or displacement applied to the top of the pile?

- Elasticity with Mohr-Coulomb plasticity in ABAQUS demand more parameters than the ones reported (dilatancy and K0). Please provide the full list of parameters.

- The authors must present references for the soil-pile friction angle values reported and its range.

- The authors must better explain the "multiple ODB import" method considered. For example, which simulation was done previously? How was it imported? Which issues did they face that prevented them from using the Geostatic option in Abaqus?

- The authors must justify mesh sizes. Besides, they indicate the values 0.11 and 0.8, but in which units? Meters?

- Did the authors perform any convergence study for the mesh sizes?

- Which type of finite element was used? Any reason to, apparently, disregard an axisymmetric simulation?

b) About the paper, in general:

- Is it really important to consider side friction? Which are the drawbacks from practical point of view? By choosing a penalty model and the correspondent friction angle, the authors are ultimately indicating how the pile shaft interacts with the soil. How does construction quality impair this?

- The authors indicate they would consider infinite elements on future works, but I am not sure this is really an issue to be dealt with, since the model size seems to be sufficiently large. Please elaborate on that.

- The whole paper gravitates around the fact that the Chinese Standard does not consider side friction when computing the bearing capacity of the piles. I assume this is derived from a practical issue, which is the varying quality of the soil-pile contact during the execution of the pile itself. Authors must elaborate on how they think different soil-pile contact conditions would make it still worth it to consider the lateral friction contribution to the bearing capacity.

- Comparing the bearing capacities obtained by the Chinese Standard and the finite element simulation should be made in two scenarios: pure base contributions (to check how the standard departs from a finite element simulation) and then adding the friction contribution. If only the base+friction case is discussed, which was the case of the paper, it is not possible to check the correctness of the standard to perform what is was supposed to do (only consider the base contribution).

- Since considering the friction contribution is deeply related to the quality of the soil-pile contact, it is important to show how different values of the penalty friction impact the final bearing capacity. This would make the readers more aware of how the Chinese Standard performs in practice.

- Overall, the authors must bring discussions and insights which are different from the ones already discussed in the works [3] and [13]. Without that, it is impossible to check the full originality of the present paper.

Reviewers' comments:

Reviewer's Responses to Questions

**Comments to the Author**

1. Is the manuscript technically sound, and do the data support the conclusions?

Reviewer #1: Yes

Reviewer #2: Yes

2. Has the statistical analysis been performed appropriately and rigorously? 

Reviewer #1: N/A

Reviewer #2: N/A

3. Have the authors made all data underlying the findings in their manuscript fully available?

Reviewer #1: No

Reviewer #2: Yes

4. Is the manuscript presented in an intelligible fashion and written in standard English?

Reviewer #1: Yes

Reviewer #2: Yes

5. Review Comments to the Author

Reviewer #1: The authors presented a study on finite element simulation of single piles with ram-compacted bearing spheres. The manuscript is finely written, however the conclusions and results are plain and nothing new found in the manuscript. Specifically,

(1) Different font sizes are used on page 12 and others, which should be avoided.

(2) All the parameters used in the numerical modeling are assumed, which makes the obtained results not convincing and practical. And the results were not validated or verified through field or lab results.

Reviewer #2: The authors conducted the numerical analysis of the bearing capacity of the single pile with ram-compacted bearing sphere. The topic of the manuscript is very interesting. Overall, this paper is well written and easy to understand. However, the reviewer would like to recommend moderate to minor revision before it can be acceptable for the publication.

The authors are advised to consider following comments in the revision.

1, The proofreading by a native English speaker for the revised manuscript is recommended.

2, The abstract should be concise and can be combined.

3, Figure 4, what is the criterion to define the zone for the dry concrete, compacted filling, compacted soil and affected soil in the model? And how does it relate to the field construction?

4, the unit of the cohesion should be kPa not Kpa.

5, It seems there is no experimental data for the verification of the numerical results.

6, Figure 9: please use different legends rather than different colors to represent ZC1 to ZC6. Similar comments for Figures 10 and 11.

7, Conclusion and Recommendation can be combined. Some non-conclusive statements can be removed.

6. PLOS authors have the option to publish the peer review history of their article (what does this mean?). If published, this will include your full peer review and any attached files.

Reviewer #1: **Yes: **Mijia Yang

Reviewer #2: No

---

## [Author Response · Author response to Decision Letter 0]

21 Jul 2023

All responses are in the 'Response to Reviewers' file

---

## [Decision Letter · Decision Letter 1]

17 Aug 2023

PONE-D-23-17951R1Finite Element Simulation Study on Vertical Bearing Characteristics of Single Pile with Ram-Compacted Bearing SpherePLOS ONE

Dear Dr. Huang,

Thank you for submitting your manuscript to PLOS ONE. After careful consideration, we feel that it has merit but does not fully meet PLOS ONE’s publication criteria as it currently stands. Therefore, we invite you to submit a revised version of the manuscript that addresses the points raised during the review process.

In order to perform the final evaluation of the manuscript, two major concerns need to be better addressed:

a) The authors presented some explanations only in the "Reply to reviewers" file, but did not incorporate those into the paper. All the issues need to be explained in the paper itself, as the doubts raised by the reviewers are also potential doubts of readers.

b) The authors indicated that they would "carefully study and evaluate the ideas and results presented in the existing literature [3] and [13], and will further deepen our discussion and analysis to ensure that they are different or more in-depth than those presented in the existing literature. We will look for new perspectives, provide new explanations, or extend existing theoretical frameworks to make our discussions unique and add them to the paper.". Such discussions were not included.

Thus, before acceptance, these two issues need to be properly addressed.

We look forward to receiving your revised manuscript.

Kind regards,

Luan Carlos de Sena Monteiro Ozelim, D.Sc.

Academic Editor

PLOS ONE

Journal Requirements:

Additional Editor Comments:

Dear authors,

Thank you for updating the paper and explaining most of the issues raised.

I still have two major concerns which need to be better addressed:

a) The authors presented some explanations only in the "Reply to reviewers" file, but did not incorporate those into the paper. All the issues need to be explained in the paper itself, as the doubts raised by the reviewers are also potential doubts of readers.

b) The authors indicated that they would "carefully study and evaluate the ideas and results presented in the existing literature [3] and [13], and will further deepen our discussion and analysis to ensure that they are different or more in-depth than those presented in the existing literature. We will look for new perspectives, provide new explanations, or extend existing theoretical frameworks to make our discussions unique and add them to the paper.". Such discussions were not included.

Thus, I advise that these two issues need to be properly addressed.

Reviewers' comments:

Reviewer's Responses to Questions

**Comments to the Author**

1. If the authors have adequately addressed your comments raised in a previous round of review and you feel that this manuscript is now acceptable for publication, you may indicate that here to bypass the “Comments to the Author” section, enter your conflict of interest statement in the “Confidential to Editor” section, and submit your "Accept" recommendation.

Reviewer #1: All comments have been addressed

Reviewer #2: All comments have been addressed

2. Is the manuscript technically sound, and do the data support the conclusions?

Reviewer #1: Partly

Reviewer #2: Yes

3. Has the statistical analysis been performed appropriately and rigorously? 

Reviewer #1: N/A

Reviewer #2: N/A

4. Have the authors made all data underlying the findings in their manuscript fully available?

Reviewer #1: No

Reviewer #2: Yes

5. Is the manuscript presented in an intelligible fashion and written in standard English?

Reviewer #1: Yes

Reviewer #2: Yes

6. Review Comments to the Author

Reviewer #1: The authors addressed most of the reviewer's comments. However, the reply to the reviewer's comments was not organized well and mixed together. It is better to separate the replies to each reviewer's comments.

Overall, the technical part of the manuscript is fine, but not new. Short of experiment verification is also a drawback, which affects the confidence of engineers to accept the reached results.

Reviewer #2: No further comments, the authors have addressed most of comments from the reviewer. The revised manuscript is ready for the publication.

7. PLOS authors have the option to publish the peer review history of their article (what does this mean?). If published, this will include your full peer review and any attached files.

Reviewer #1: No

Reviewer #2: No

---

## [Author Response · Author response to Decision Letter 1]

24 Aug 2023

The answers to all questions can be found in their respective 'response to reviewer' files

---

## [Editor Report · Decision Letter 2]

6 Sep 2023

Finite Element Simulation Study on Vertical Bearing Characteristics of Single Pile with Ram-Compacted Bearing Sphere

PONE-D-23-17951R2

Dear Dr. Huang,

We’re pleased to inform you that your manuscript has been judged scientifically suitable for publication and will be formally accepted for publication once it meets all outstanding technical requirements.

Kind regards,

Luan Carlos de Sena Monteiro Ozelim, D.Sc.

Academic Editor

PLOS ONE

Additional Editor Comments (optional):

The authors have properly addressed all the issues raised. Therefore, the paper can be accepted.
---

## [Editor Report · Acceptance letter]

12 Sep 2023

PONE-D-23-17951R2 

Finite Element Simulation Study on Vertical Bearing Characteristics of Single Pile with Ram-Compacted Bearing Sphere 

Dear Dr. Huang:

I'm pleased to inform you that your manuscript has been deemed suitable for publication in PLOS ONE. Congratulations! Your manuscript is now with our production department. 

Kind regards, 

on behalf of

Dr. Luan Carlos de Sena Monteiro Ozelim 

Academic Editor

PLOS ONE